# Light Scattering as an Easy Tool to Measure Vesicles Weight Concentration

**DOI:** 10.3390/membranes10090222

**Published:** 2020-09-03

**Authors:** Giulia Di Prima, Fabio Librizzi, Rita Carrotta

**Affiliations:** 1Institute of Biophysics, National Research Council, Via Ugo La Malfa 153, 90146 Palermo, Italy; giulia.diprima@virgilio.it (G.D.P.); fabio.librizzi@pa.ibf.cnr.it (F.L.); 2Dipartimento di Scienze e Tecnologie Biologiche Chimiche e Farmaceutiche (STEBICEF), University of Palermo, Via Archirafi 32, 90123 Palermo, Italy

**Keywords:** vesicles, extrusion, Stewart assay, weight concentration, light scattering, spectrofluorimeter

## Abstract

Over the last few decades, liposomes have emerged as promising drug delivery systems and effective membrane models for studying biophysical and biological processes. For all applications, knowing their concentration after preparation is crucial. Thus, the development of methods for easily controlling vesicles concentration would be of great utility. A new assay is presented here, based on a suitable analysis of light scattering intensity from liposome dispersions. The method, tested for extrusion preparations, is precise, easy, fast, non-destructive and uses a tiny amount of sample. Furthermore, the scattering intensity can be measured indifferently at different angles, or even by using the elastic band obtained from a standard spectrofluorimeter. To validate the method, the measured concentrations of vesicles of different matrix compositions and sizes, measured by light scattering with different angles and instruments, were compared to the data obtained by the standard Stewart assay. Consistent results were obtained. The light scattering assay is based on the assessment of the mass fraction lost in the preparation, and can be applied for methods such as extrusion, homogenization, French press and other microfluidic procedures.

## 1. Introduction

Liposomes are lipid systems that phospholipids form in aqueous solutions. Due to their easy preparation, composition and versatility, they have received considerable attention as great potential pharmaceutical carriers and as effective membrane models over the past 30 years [1]. Unilamellar liposomes, called vesicles, constitute a more reliable model of the typical lipid structure of a membrane, isolating a relatively large hydrophilic space from the bulk through a single thin hydrophobic lipid bilayer. They can be prepared in a variety of sizes, ranging from tens of nanometres to tens of micrometres, thus emulating different kinds of cell compartments or organelles. Various types and mixtures of lipids can be used to prepare the liposomes [2,3,4,5].

As drug delivery systems, liposomes have emerged as one of the most promising tools for drug targeting in different medical fields because of their inherent and peculiar characteristics, such as biocompatibility, biodegradability and low toxicity, coupled in the case of vesicles with the ability to encapsulate both hydrophobic and hydrophilic molecules. Moreover, great biotechnological advances have led to the widespread use of vesicles in diverse areas, ranging from therapeutics to diagnostics and, more recently, theranostic applications [6,7,8,9,10,11,12,13]. On the other hand, lipid vesicles are the most popular model membrane systems for studying biological issues [14,15,16,17].

Liposomes can also be used to model physiological extracellular vesicles, called exosomes, which nowadays are receiving great attention for their involvement in cell–cell communication and their potential to be converted into biocompatible drug delivery systems, after suitable bioengineering [18,19]. Modeling natural vesicles with a suitable mixing of unilamellar liposomes allows us to obtain an approximate, but experimentally very useful, estimate of their concentration. For all these applications, controlling sample concentration is absolutely necessary in order to (i) accurately replicate experimental conditions, (ii) study properties which depend on nanoparticle amount, (iii) study properties which depend on the ratios of various components of the solution, (iv) evaluate any material loss during the sample preparation steps (e.g., extrusion, homogenization, etc.), and (v) model more complex systems.

The Stewart assay is one of the traditional methods for the determination of phospholipid concentration, and it is often used after liposome preparation. The assay is based on the ability of phospholipids to form a colored complex with ammonium ferrothiocyanate, whose light absorption is converted into a weight-amount of phospholipids [20,21]. This method presents the following limitations: (i) it is insensitive to phosphatidylglycerol-based phospholipids, as well as other molecules present in the bilayers, such as cholesterol or gangliosides, so that in multicomponent samples with unknown compositions it detects only a generic phospholipids presence; (ii) it destroys the sample; (iii) it requires the use of organic solvents and different chemicals; and (iv) it depends upon a calibration curve, which requires exact knowledge of the lipid matrix composition. In the past, beside the Stewart assay, other methods have been used to control lipid concentration, such as the phosphorous Bartlett assay or the enzymatic assay to determine the choline group by a PC-specific phospholipase [22,23]. The first is based on a colorimetric test after a several-step reaction, which is quite complicated and time consuming, to detect the total phosphorus amount, attention being paid to the phosphorus present not in phospholipids. The second method, which is more manageable, is specifically used to quantify the amount of the main lipid component present in biological environments, the phosphatidylcholine, this being intrinsically limited. A colorimetric assay is used to determine the choline concentration via a combined enzymatic method, by using phospholipase D, choline oxidase and peroxidase.

Nowadays, finding novel and effective alternative methods to control liposome concentrations is an actual scientific research goal. A recent study has proposed an NMR-based method to determine liposome concentration [24]. Access to NMR instrumentation, however, is not straightforward. As a consequence, finding alternative methods to measure vesicles concentration, which can be performed by using accessible and common laboratory facilities, is still an actual issue.

Based on these considerations, this work proposes a novel method to evaluate small, large and giant vesicles (SUV, LUV and GUV) weight concentrations by static light scattering analysis, right after their preparation. The method is based on the relation between the static scattering intensity, the mass concentration and the average molar mass of the vesicles in solution. It can be applied at any stage in the preparation, and the total lipid mass can be controlled and almost completely retrieved by means of a second step of rinsing, like in the case of extrusion, homogenization, French press or other microfluidic setups. The assay is described here by using vesicles prepared with the extrusion method. The proposed approach shows advantages of being easy, non-destructive and fast. Moreover, it does not imply the use of organic solvents or other chemicals, and it is sensitive to the whole mass, including all the lipid components present in the nanosystem, proteins or drugs, if present. Then, a mass control of the sample could be obtained even though the matrix components are not well specified, like in the case of asolectin, a mixture of lipids often used to prepare liposomes for many different uses, such as for encapsulating active biomolecules [25]. Furthermore, to be applied, the method simply requires a reliable measurement of the intensity of the light scattered by vesicle dispersions, which can be achieved, of course, by using light scattering devices, but also by using, in an unconventional way, a spectrofluorimeter. In fact, standard spectrofluorimeters, commonly present in many laboratories, allow the detection of the elastic band, i.e., the band obtained by setting equal values for excitation and emission wavelengths. Regardless of the fluorescence properties of a sample, this band actually gives a reliable estimation of the intensity of the light scattered by the sample [26]. Results obtained by the LS-based method, at different scattering angles and with different instruments, demonstrate methodological strength. Furthermore, the Stewart assay is taken as the reference method to validate the LS-based one. Comparison of the results shows consistency, and leads therefore to validation.

## 2. Materials and Methods

### 2.1. Materials

The solvents and chemicals were of analytical grade, and were used without any further purification. 2-Oleoyl-1-palmitoyl-sn-glycero-3-phosphocholine (POPC) was purchased from Avanti Polar Lipids (Avanti Polar Lipids Inc., Alabaster, AL, USA), INC. 2-Oleoyl-1-palmitoyl-sn-glycero-3-phospho-L-serine sodium salt (POPS), Cholesterol, Chloroform, FeCl_3_ and NH_4_SCN (used to prepare the ammonium ferrothiocyanate solution for the Stewart Assay) were purchased from Sigma-Aldrich (Merck Life Science S.r.l., Milano, Italy).

### 2.2. Liposomes Preparation

Unilamellar vesicles (UVs) were obtained following a multistep procedure. Lipid standard solutions (1 mg/mL) were prepared by dissolving each lipid (POPC, POPS and Cholesterol) in chloroform. Subsequently, aliquots of the standard solutions were mixed in the appropriate ratio. Afterwards, chloroform was evaporated under a gentle nitrogen flux in order to obtain lipid films with a total lipid mass of 1 mg. Multi lamellar vesicles (MLVs) were initially obtained by hydration of the prepared lipid films with bidistilled water (1 mL). Then, 5 freezing and thawing (FAT) cycles were performed in order to recover all the lipid material from the glass and incorporate it in the MLVs. The obtained dispersions were extruded through to a mini-Extruder Avestin (31 passes, with 50, 100 or 200 nm polycarbonate filters). To achieve a complete recovery of the whole lipid mass from the extrusion apparatus, a second extrusion cycle was performed by adding fresh bidistilled water. Finally, 2 samples were recovered from the 2 extrusion cycles (extrusion I and extrusion II). This double procedure is crucial to evaluate liposomes mass concentration by light scattering (see below).

Two different lipid matrix were studied, one composed of 100% *w*/*w* POPC, and one composed of POPC 85.5% *w*/*w*, POPS 9.5% *w*/*w* and Cholesterol 5% *w*/*w*. Three different size vesicles (50 nm, 100 nm and 200 nm) were prepared for each matrix. The different UVs are indicated as POPC_LIPO_50, POPC_LIPO_100 and POPC_LIPO_200, and MIX_LIPO_50, MIX_LIPO_100 and MIX_LIPO_200. Each sample was prepared in duplicate and analyzed in triplicate.

### 2.3. Stewart Assay

The Stewart assay is based on the ability of phospholipids to form a colored complex with ammonium ferrothiocyanate in organic solution. Indeed, the experimental set up needs 2 different phases. The aqueous phase consists of 2 mL of 0.1 M ammonium ferrothiocyanate solution (prepared by dissolving 27.03 g/L of FeCl_3_ and 30.40 g/L of NH_4_SCN in bidistilled water), while the organic phase consists of 2 mL of chloroform. These 2 phases are vortexed for at least 15 s and then left until complete separation for 2 h. Afterwards, the organic phase is recovered and the absorbance at 485 nm is measured. To construct the calibration curve, the organic phase consists of lipid standard solutions in chloroform (total lipid concentration range: 0.005–0.050 mg/mL), the concentration of which is established by weighing the lipids dissolved in a mother chloroform solution. It is worthy to note that, to obtain the calibration curve, the exact composition of the lipid mixture is needed (standards for POPC_LIPO evaluation are composed by POPC, while standards for MIX_LIPO evaluation are composed by POPC:POPS:Chol=85.5:9.5:5.0). To evaluate sample concentration, 50 μL of aqueous UVs dispersion were added to 2 mL of chloroform, briefly vortex and left for 1 h. Subsequently, 2 mL of ammonium ferrothiocyanate solution were added and the binary mixture was vortexed again and then left for 2 h until complete separation. Each experiment here reported was performed in triplicate.

### 2.4. Light Scattering Assay to Evaluate Mass Concentration

Samples for light scattering analysis (LS) were prepared by the dilution of the samples obtained in liposome preparation after the two extrusion steps (extrusion I and extrusion II). When not differently specified, the samples, prepared by dissolving about 1 mg/mL of lipids, were respectively diluted 1:50 (extrusion I) and 1:5 (extrusion II) with fresh bidistilled water, placed into a dust-free quartz cell without further filtering and kept at 20 ± 0.1 °C in the thermostatic cell compartment of a Brookhaven Instruments BI200-SM goniometer. The static scattered light intensity and the intensity autocorrelation function were measured by using a Brookhaven BI-9000 correlator and a 100 mW solid-state laser (λ = 532 nm) at 3 different scattering angles: 60°, 90° and 120°. Dynamic light scattering (DLS) was used to characterize the hydrodynamic size of the vesicles and to confirm the goodness of the samples obtained in both the extrusion steps (I and II). Samples were characterized by a CONTIN-like analysis and a cumulants analysis. According to the cumulants analysis, the *g*^(2)^(*t*) can be expressed as follows [27]:(1)$$g^{\left(2\right)}\left(t\right)=B+\beta \exp\left(-2\bar{\Gamma }t\right){\left(1+\frac{\mu_{2}}{2!}t^{2}+\ldots \right)}^{2}$$

This analysis gives as a first approximation $\bar{\Gamma }$, and the second moment (*μ*_2_) of the distribution around $\bar{\Gamma }$. The $\bar{\Gamma }$ value through the Stocks–Einstein relation, $\bar{\Gamma }=\frac{KTq^{2}}{3\pi \eta {\bar{D}}_{h}}$, is related to the average hydrodynamic diameter (${\bar{D}}_{h}$), with *K* as the Boltzmann constant, *T* the temperature, *q* the scattering vector (*q* = 22.3 μm^−1^ for *θ* = 90°, since $q=\frac{4\pi }{\lambda }\sin\frac{\theta }{2}$) and *η* the solvent viscosity.

The CONTIN-like analysis allows us to extract the full size distribution *G*(Γ), and thus to compare the two extrusion samples. In this case, *g*^(2)^(*t*) is expressed by a Laplace transformation, as follows:(2)$$g^{\left(2\right)}\left(t\right)=B+\beta {\left[\int_{0}^{\infty }G\left(\Gamma \right)e^{-\Gamma t}d\Gamma \right]}^{2}$$

Light scattering measurements were used to calculate sample mass concentration, as explained in the Results and Discussion section, by considering for both extrusion I and extrusion II samples the following parameters: the actual scattered light intensity (*I*_1_ and *I*_2_), obtained after buffer scattering subtraction; sample dilution (*d*_1_ and *d*_2_); extruded volume (*V*_1_ and *V*_2_); starting total mass of lipids (*M*_0_) and unrecovered or dead volume (Δ*V*). It is worth noting that dilutions are chosen depending on the roughly expected concentrations, the LS setup sensitivity and the set experimental parameters, such as the incident power and pin hole of detection. It is essential that the two samples (I and II) are measured in the same conditions. Each experiment here reported was performed in triplicate.

### 2.5. Spectrofluorimetric Analysis to Evaluate Mass Concentration

The Brookhaven LS measurements were complemented by measuring the area underneath the elastic peak obtained in the fluorescence spectrum of the vesicle dispersions. In fact, when particles are dissolved in a solution and hit by radiation, they produce elastic scattering radiation in all directions. Like in a traditional light scattering setup, the intensity of the elastically diffused light will be proportional to the mass of the particles and to their concentration. The elastically diffused light can be measured by using a spectrofluorimeter, which is a quite common spectroscopy lab facility, by simply setting equal values for the excitation and the emission wavelength [26]. Therefore, the same samples and dilutions used in the Brookhaven setup were used in a JASCO FP-6500 spectrofluorimeter, and the area underneath the elastic peak obtained with a 532 nm excitation wavelength, 1 nm excitation and emission slits and 500 V PMT voltage was taken as a measure of the light scattered by vesicle dispersions. Each experiment was performed in triplicate.

### 2.6. Statistics and Errors Calculation

The errors associated with the concentration values were obtained by a statistical approach. For both lipid matrices, six independent films, although coming from the same weighing event, were deposed and, after film hydration, two liposome preparations were obtained for each of the filter dimensions (50, 100 and 200 nm) by independent extrusion procedures. Thus, for each independent extrusion procedure, three dilutions of the liposome dispersion were analyzed so as to obtain a Statistical Error (SE) to be assigned to the average measured value (SE is calculated as the error on the average). For the Stewart assay, this value (i.e., the average Abs ± SE) leads to the concentration through the calibration curve, the concentration error being determined by error propagation, $\varepsilon_{\exp}=f_{Stew}\left(SE\right)$. For the LS assay, the two measured values (*I*_1_ and *I*_2_ ± SE), corresponding to extrusion I and II, lead to the concentration via the formula reported in the theory section, the error associated calculated by error propagation, $\varepsilon_{\exp}=f_{LS}\left(SE\right)$. In both cases, for Stewart and LS, and for each liposome size, a unique average was obtained, calculated from the two average values corresponding to the different preparations. To this value two errors were associated, one coming from the propagation of the experimental errors, obtained for each single preparation (*A* and *B*) and reflecting the uncertainty of the measure ($\varepsilon_{AB}=\frac{\sqrt{\varepsilon_{\exp A}^{2}+\varepsilon_{\exp B}^{2}}}{2}$), and one due to the different preparations and their discrepancy, reflecting the uncertainty of the preparation procedure ($\sigma_{AB}=\sqrt{\Delta x_{A}^{2}+\Delta x_{B}^{2}}$; where $\Delta x=x-{\bar{x}}_{A,B}$). The maximum between these two was attributed to the final average value. This approach was necessary because in some cases the errors associated with the values obtained in the single preparation by the LS-based method were very small, with a consequent inconsistency among different measurements, but this still being always consistent with the Stewart assay determination. In fact, due to the high sensitivity of the light scattering technique to tiny differences in the repeated samples, especially for the multi-angle measurements, a consideration of different preparations can better describe the variability of samples. The same rationale was adopted both for the LS and Stewart final uncertainties.

## 3. Results and Discussion

Generally, the preparation of unilamellar liposomes requires previously-prepared multi lamellar vesicles (MLVs), which can be subjected to a variety of different procedures (e.g., extrusion, homogenization, sonication, ethanol diffusion, detergent dialysis, etc.) in order to opportunely modulate their dimension and lamellarity [28]. In this work, MLVs were prepared by Thin Layer Evaporation (TLE) followed by freezing and thawing (FAT) cycles. Afterword, UVs were obtained by extrusion through polycarbonate filters, following a well-known multistep procedure [25]. Importantly, the method here proposed for the determination of concentration needs two different and sequential extrusion steps, as follows: (I) the MLV solution, containing the lipid mass *M*_0_, is loaded into the syringe and the setup is sealed to proceed to the first extrusion (including 31 steps), ending with the recovery of the extruded volume, *V*_1_; (II) a volume of solvent is then added into the loading syringe, taking care to not lose the sample and after sealing the setup again, the second extrusion (a few steps are needed) is performed, which gives the extruded volume *V*_2_. This second step also has the ability to recover almost all the lipid mass remaining in the setup after the main extrusion, in a second sample. This will be diluted with respect to the main one, by as much as is the amount of added buffer between the two extrusion steps.

### 3.1. Theory

#### 3.1.1. LS-Based Method

During the procedure it is essential to fix the following:The weighted total mass of lipid components employed for the preparation of the starting film (*M*_0_);The recovered volume after the first extrusion cycle, *V*_1_;The recovered volume after the second extrusion cycle, *V*_2_;The total volume inserted into the extruder, *V*_T_, i.e., the sum of the starting volume injected and the volume of solvent which is added to perform the second extrusion and rinse the setup. The total volume *V*_T_ does not correspond to *V*_1_ + *V*_2_, since some volume, the dead volume Δ*V*, remains trapped into the setup.

The volume obtained from the second extrusion can be measured, and this will allow us to determine the dead volume of the setup, Δ*V*, containing some residual lipid mass. The volumes can be measured through the syringe scale (as is done in the experiments reported here), or by weighing the solutions and calculating the corresponding volumes (assuming a density ρ = 1 g/mL).

It is worth noting that information on the initial mass of lipids in the preparation (*M*_0_) and on the obtained sample volumes does not straightforwardly give the weight concentration of the sample, exactly as in the case of any biological sample preparation. For instance, when preparing a protein solution, the knowledge of the initial protein weight and of solvent volume does not immediately give the protein concentration, which in fact is usually measured spectrophotometrically, to take into account any mass loss and/or experimental artefact during the various steps of sample preparation (weighing, filtration, sample pouring, etc.). In all these cases, an evaluation of the final weight concentration of the sample is always required.

Light scattering intensity on appropriate dilutions of the two extruded solution (I and II) is measured, in order to calculate the corresponding weight concentrations. The static light scattering intensity, detected at the angle *θ*, depends on the weight averaged molecular mass (*M_w_*), the weight concentration (*C*) and the so-called form factor (*P**_θ_*), as follows [29,30]:(3)$$I=kM_{w}CP_{\theta }$$

Here, *k* is a constant depending on the experimental setup and sample parameters, such as the refractive index of the solvent (*n*_0_) and the derivative of the differential solute refractive index, with respect to the solute concentration d*n*/d*C*, the parameter giving the so called contrast in light scattering. Equation (3) is true in diluted conditions, i.e., in the absence of particle interactions, as in the case of the solutions here studied. *P**_θ_* can be approximated to 1 when the solute particle dimension *D* is smaller than *λ*/10.

For the two solutions, I and II, considering that the light scattering is measured on samples diluted by a factor *d*_1_ and *d*_2_ respectively, we obtain the following equations:(4)$$I_{1}d_{1}=kM_{w}C_{1}P_{\theta }$$
(5)$$I_{2}d_{2}=kM_{w}C_{2}P_{\theta }$$

The constants *k* and *M_w_* are the same for both samples, as well as the form factor *P**_θ_*. In fact, both *M_w_*, the weight-averaged molecular mass, and *P**_θ_*, the form factor, depend on the vesicle size distribution, which is equal in the two samples (as shown below in the Dynamic Light Scattering section).

Equations (2) and (3) can be combined as follows:(6)$$\frac{I_{1}d_{1}}{I_{2}d_{2}}=\frac{C_{1}}{C_{2}}$$

Equation (6) has two unknown variables, which are *C*_1_ and *C*_2_. However, considering the previously described extrusion parameters, an equation for the conservation of the mass can be written as
(7)$$M_{0}=C_{1}V_{1}+C_{2}\left(V_{T}-V_{1}\right)$$

It is worthy of note that it is assumed here that *V*_2_ and the dead volume Δ*V* solutions have the same concentration, *C*_2_. This assumption, which is quite plausible, considers as negligible the amount of spare lipids trapped in the extruder but not included in liposomes. Clearly, $V_{2}+\Delta V=V_{T}-V_{1}$.

Consequently, combining Equations (6) and (7) gives:(8)$$\frac{M_{0}}{V_{1}}-\frac{C_{2}\left(V_{T}-V_{1}\right)}{V_{1}}=C_{2}\frac{I_{1}d_{1}}{I_{2}d_{2}}$$

To simplify, by posing
$$\begin{array}{l} \alpha =\frac{M_{0}}{V_{1}}\\ \upsilon =\frac{V_{T}-V_{1}}{V_{1}}\\ \gamma =\frac{I_{1}d_{1}}{I_{2}d_{2}} \end{array}$$
it is possible to rewrite Equation (8) and easily calculate both *C*_1_ and *C*_2_ as follows:(9)$$\begin{array}{l} C_{2}=\frac{\alpha }{\gamma +\upsilon }\\ C_{1}=\alpha -\upsilon C_{2}=\frac{\alpha \gamma }{\gamma +\upsilon } \end{array}$$

The method can apply to vesicles of whatever dimensions, and the signal can be collected at any scattering angle for the two samples. In fact, as said before, due to the consistent identity of the vesicle size distribution for the two extrusions, both mass and form factor contributions are negated.

It has to be noticed that the limit of detection of the concentration depends on the measure of the intensity (both *I*_1_ and *I*_2_), which depends on the experimental setup used, the average molecular mass, the effective sample concentration and the vesicle contrast. By modeling the vesicle as a sphere with all the lipid mass concentrated in a shell of known diameter *D*, one can assume that *M* is proportional to *D*^2^, with $M\left(D\right)/D^{2}\approx 2\pi M_{lip}/\Delta A_{lip}$, where *M*(*D*) is the mass of a vesicle with diameter *D*, *M_lip_* is the mass of a single phospholipid, and Δ*A_lip_* is the average area occupied by the lipid head group in the vesicle (for POPC, Δ*A_PCp_* = 0.63 nm^2^) [31]. For *D* = 50 nm, the concentration limit of the measurement is about 0.01 mg/mL (at 90°, with a 100 μm pin hole and a power laser of 15 mW), for *D* = 100 nm it is about 4 times smaller, and reduces to 16 times smaller for 200 nm vesicles. By increasing the pin hole one can further improve the number of detected photons (although the dynamical characterization gets worse), and therefore achieve even greater concentration sensitivity.

Similarly, the scattering intensity of a vesicle dispersion can be measured using a spectrofluorimeter, simply by setting equal values for the excitation and emission wavelength [26]. Then, *I*_1_ and *I*_2_ values, and thus γ, can be evaluated by measuring the scattering intensity from the areas underneath the elastic peak in a fluorescence emission spectrum, after the subtraction of the background. In this case, the emission spectrum with a chosen excitation wavelength (as an example, here λ_ex_ = 532 nm) is measured for the two samples I and II, and the elastic peak is taken into account, that is, the peak centered at the same wavelength of the excitation beam (@λ_em_ = λ_ex_ = 532 nm). For a better signal to noise ratio, the area values *A*_1_ and *A*_2_ underneath the elastic Rayleigh band, relative to the extrusion I and II, respectively, are calculated and used in Equation (6) as a measure of *I*_1_ and *I*_2_.

A specific section, reporting the detailed step-by-step analysis procedure for both a representative single extrusion sample (one POPC_LIPO_50 sample) and samples from two independent extrusion processes (POPC_LIPO_100_A and POPC_LIPO_100_B samples), is included in the Supporting Materials.

#### 3.1.2. Samples Characterization by Dynamic Light Scattering (DLS)

As previously mentioned, samples were characterized by DLS in order to measure the hydrodynamic size distribution of the unilamellar vesicle dispersion, to confirm both the goodness of the sample in terms of homogeneity and to demonstrate the similarity of the two extrusions samples, in terms of weight averaged mass (*M_w_*) and vesicle size distribution, or form factor (*P**_θ_*).

In Figure 1, the intensity autocorrelation functions data for two representative (POPC_LIPO_50 samples) extruded solutions I and II are reported, together with the fitting function arising from a cumulants analysis and from a CONTIN-like method [27,32].

The cumulants analysis gives $\bar{\Gamma }\left(I\right)=\left(0.00240\pm 0.00002\right){\mu \sec}^{-1}$ (${\bar{D}}_{cum}=\left(90\pm 1\right)\mathrm{nm}$) and $\bar{\Gamma }\left(\mathrm{II}\right)=\left(0.00260\pm 0.00002\right){\mu \sec}^{-1}$ (${\bar{D}}_{cum}=\left(83\pm 1\right)\mathrm{nm}$), and for the distribution widths, $\mu_{2}(I)=\left(5.5\pm 1.1\right)\times 10^{-7}{\mu \sec}^{-2}$ and $\mu_{2}(\mathrm{II})=\left(4.4\pm 0.8\right)\cdot 10^{-7}{\mu \sec}^{-2}$, respectively. The second extrusion appears to be slightly more homogeneous, as reflected in the distribution width, probably due to the additional passes executed. However, as is evident from the experimental data reported in Figure 1, the results corresponding to the different samples, after suitable normalization, are almost indistinguishable, although the analysis can detect some minimal difference, as also evidenced in the intensity weighted size distribution obtained by the CONTIN-like analysis (B). From that it is possible to calculate the harmonic average hydrodynamic diameter, obtaining for the $G{\left(D_{h}\right)}_{I}$
${\bar{D}}_{harm}=82\;\mathrm{nm}$ and for $G{\left(D_{h}\right)}_{\mathrm{II}}$
${\bar{D}}_{harm}=80\;\mathrm{nm}$. The fact that the samples obtained by the two extrusions offer identical DLS analysis results confirms that their form factor, important for the static light scattering measurement, can be assumed to be equal (see above). Similar results were obtained for all the other samples prepared and analyzed in the present study. Concerning the samples obtained with 100 nm and 200 nm filters, as an example, for the main extrusion of the MIX_LIPO sample, hydrodynamic diameters of *D_cum_* = (101 ± 2) nm and (199 ± 3) nm were obtained by cumulants analysis, respectively. For all samples, an estimation of the polydispersion index (PI = $\mu_{2}/{\bar{\Gamma }}^{2}$) gives PI ≈ 0.06.

### 3.2. Method Validation

To validate the here-proposed method, the POPC_LIPO and MIX_LIPO weight concentrations were evaluated by the light scattering method (both employing an LS instrument and a spectrofluorimeter) and compared to values obtained by performing the traditional Stewart assay for all the samples.

It is important to point out that the compared values are those referring to the first extrusion (samples with higher concentration). Indeed, the second extrusion samples just contain the residual amounts of lipids, but they are crucial to performing the LS-based method.

First, to execute the Stewart assay, it is necessary to construct the appropriate calibration curve. In this work, two different calibration curves were evaluated: simple liposomes composed of POPC (POPC_LIPO), and mixed liposomes composed of POPC, POPS and cholesterol (MIX_LIPO). The effectiveness of the measurement is strictly connected to the employment of an accurate lipid mix in the construction of the calibration curve, as different types of phospholipids react differently, and certain molecules such as cholesterol do not react at all.

In Figure 2, the calibration curves obtained are reported, as well as the concentration values for POPC_LIPO_50 and MIX_LIPO_50 (reported as representative samples) obtained by employing the Stewart assay.

In order to definitely validate the LS-based method, when using the LS instrument, sample concentrations were calculated by analyzing the scattering intensity data obtained at three different angles (60°, 90° and 120°). Finally, the same samples (with the same dilutions) were measured by using the spectrofluorimeter. In this case, the elastic peak corresponding to the excitation wavelength (532 nm) was monitored.

It is worth of noting that the value of intensity measured in all the cases for the two extrusions depends on the dilution applied to the original sample. This parameter is chosen to have a good signal to noise ratio in the measure of the intensity for the two samples, and it depends on the experimental parameters. Thus, the relative value of *I*_1_ and *I*_2_ alone is not meaningful; what matters is instead the product *I*·*d*. In fact, this product gives the excess scattering of the undiluted solution. Dilution is necessary since a vesicles dispersion of ~1 mg/mL concentration would saturate the scattering signal in both instruments, unless a very strong filter is applied for the incident light; this would lead to a strong reduction in the signal to noise ratio. Moreover, 1 mg/mL concentrations and even more diluted sample concentrations guarantee the absence of structure factor effects due to interparticle interactions. All the obtained data, corresponding to the different samples, are reported in Appendix A. As discussed in the previous section, the final values come from averaging the results from two independent preparations for each sample. Averaged data are reported in Figure 3, in order to compare results obtained by the different methods (Stewart assay and light scattering). In the latter case, the results derived from measuring at different angles (60°, 90° and 120°) and by using the elastic peak of the emission spectrum are all reported. Data analysis shows how the light scattering-based method gives consistent results. In particular, in all cases, the weight concentration values calculated by the novel LS-method resulted in a concentration range (concentration value ± error) corresponding to that determined by the Stewart assay; different angles results are equivalent and consistent also with the elastic peak measure on the spectrofluorimeter.

Errors of different magnitude for the different samples may arise from the small number of repetitions, this being related to more similar or dissimilar preparations.

The results makes it valid to perform this assay by using common bench instruments, such as a spectrofluorimeter or a back scattering size analyzer, often present in biotech labs.

The light scattering-based method has been thought to rigorously determine the concentration of a preparation when the mass closure relation can be applied and two dispersions with the same properties can be compared. Furthermore, it is important to notice that it gives a weight concentration value for the all components in the nanosystem, since its sensitivity is to the particle mass. In the case of home-made liposomes, their content is known. Thus, the molar concentration can also be extracted from the weight concentration. Differently, the Stewart assay gives directly the molar concentration of the reactive group in the phospholipids, as long as the phospholipids components are known, in order to use the right calibration curve.

It is worth noticing that by using the light scattering-based method, it can be possible to evaluate the unknown concentration of a repetition sample, done for example without knowing the volumes in the preparation, but knowing the composition and size. In fact, the concentration is obtained by measuring the Rayleigh ratio of the sample (*R*) and knowing the Rayleigh ratio of the corresponding standard (*R_std_*), which is a sample with the same characteristics that at least once has been measured and its concentration determined with the LS method, right after preparation. The Rayleigh ratio is in fact an absolute measurement of the scattering light, free of dependency on the experimental setup (setup geometry, incident laser power, photo multiplier sensitivity), expressed in the absolute scale in cm^−1^ [30]. A standard sample is then characterized via specific composition and obtained by extrusion through a known cut-off filter. The measure of both excess absolute intensity (*R_std_*) and weight concentration (*C_std_*) for the standard would allow one to obtain the relationship between the unknown sample’s concentration and the obtained standard parameters:(10)$$C=\frac{C_{std}}{d_{std}}d\frac{R}{R_{std}}$$
where *R* and *R_std_* are the Rayleigh ratio at the selected scattering angle, *d* and *d_std_* are the dilution factors and *C* and *C_std_* are the concentrations of the unknown sample and the standard, respectively. In addition, the simultaneous dynamic light scattering measurement offers quality control, as well as determining the size distribution of the unknown sample and the goodness of the comparison to the standard one. In the case of the here-presented LS-based assay, the Rayleigh ratio (*R*) is not strictly necessary, since, being a ratio, *I*_1_/*I*_2_ is per se free from experimental setup dependencies and *I*_1_/*I*_2_ = *R*_1_/*R*_2_. However, the intensity *I* should be converted into absolute Rayleigh ratio, if the sample is supposed to be used as a standard for size and composition by referring to its Rayleigh ratio, concentration and mass, in order to directly determine the concentration of the future preparations.

The LS-based method can be applied also when dealing with liposomes encapsulating both hydrophilic or lipophilic molecules, as well as amphiphilic ones, at least when the molecule concentration regime assures solubility in lipids or in the water environment [25]. In fact, for hydrophilic molecules, only the background may slightly change, as the molecules present as a cosolute inside and outside the vesicles; at difference, lipophilic molecules are trapped in the lipid bilayer and the method can be applied by considering also the contribution to the weight concentration coming from the loaded molecules. If needed, the evaluation of the loaded material in the bilayer provides the concentration of the lipid component alone. In the amphiphilic case, a scheme with a combination of the two conditions can be applied.

## 4. Conclusions

Unilamellar vesicles are nowadays being extensively investigated for their potential applications in many different fields of biophysical and biotechnological research [1]. In all applications, spanning from drug delivery to studies on the interactions of specific molecules with cellular membranes, for which vesicles constitute an extremely versatile and reliable model, the knowledge of their weight concentration is crucial [8,9,10,11,12,13,14,15,16,17]. As an example, any consideration of stoichiometric affinities cannot be separated from a determination of the total amount of lipids in loading vesicles [33]. As a matter of fact, however, a routine determination of the weight concentrations of vesicles is not always performed, since standard methods may require considerable lab time. Often, the initial amount of lipids used in the preparation may be used as an indication of the weight concentration, thus assuming that the loss of materials during the various steps of sample preparation (extrusion, size exclusion chromatography, etc.) could be neglected [34].

In this scenario, a simple and easy method for the determination of the weight concentration of vesicles can be of great utility for many different applications. In fact, an NMR-based method has recently been proposed as a fast alternative to standard assays [24]. However, though being a highly powerful approach, NMR instrumentation is not easily accessible in many cases.

In this work, a novel and easy method for the evaluation of vesicles’ weight concentration has been introduced and validated by comparison with a common chemical approach (Stewart Assay). The novelty is based on the light scattering quantification of the whole lipid mass, including the fraction lost during the preparation procedure. The technique can be used for preparation protocols, such as extrusion, homogenization, French press and other microfluidic methods. The approach has the advantages of (i) being fast and easy, (ii) not requiring organic solvents, (iii) evaluating any mass loss in the setup, and (iv) having a possible implementation by using a simple spectrofluorimeter or a bench instrument, such as a size analyzer. Furthermore, a full use of the light scattering technique allows one to check the preparation quality by complementing dynamic light scattering characterization, and, last but not least, to determine the concentration of a known sample (in terms of composition and size) by using the Rayleigh ratio of a corresponding standard previously prepared and characterized with the LS-based assay.

## Figures and Tables

**Figure 1 membranes-10-00222-f001:**
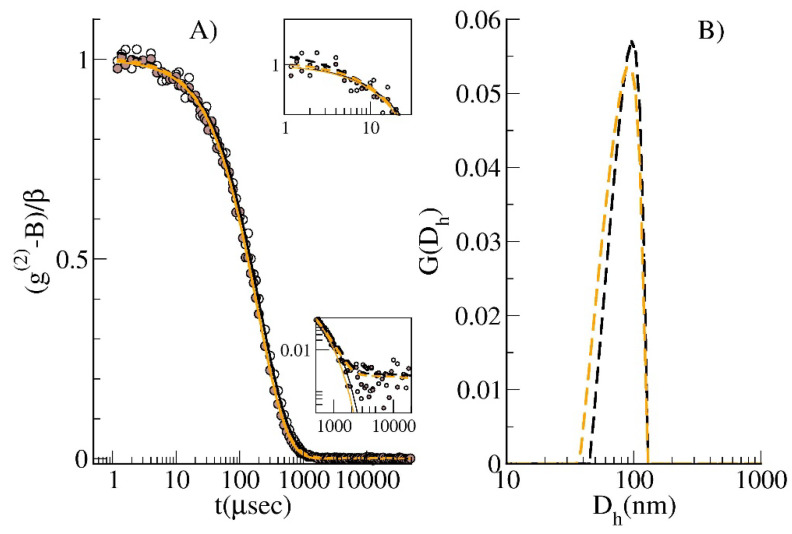
DLS analysis at θ = 90° for POPC_LIPO_50, as a representative sample. (**A**) intensity autocorrelation function, (*g*^(2)^−*B*)/*β* for extrusion I (empty circle) and II (brown circle); lines represent the fitting curves according to the cumulants (continuous lines) and contin analysis (dashed lines). Fit difference is evidenced in the insets in a log–log scale. Upper inset: zoom of low t range. Lower inset: zoom of high t-range. (**B**) Intensity weighted size distribution arising from contin analysis: extrusion I (black dashed) and II (orange dashed).

**Figure 2 membranes-10-00222-f002:**
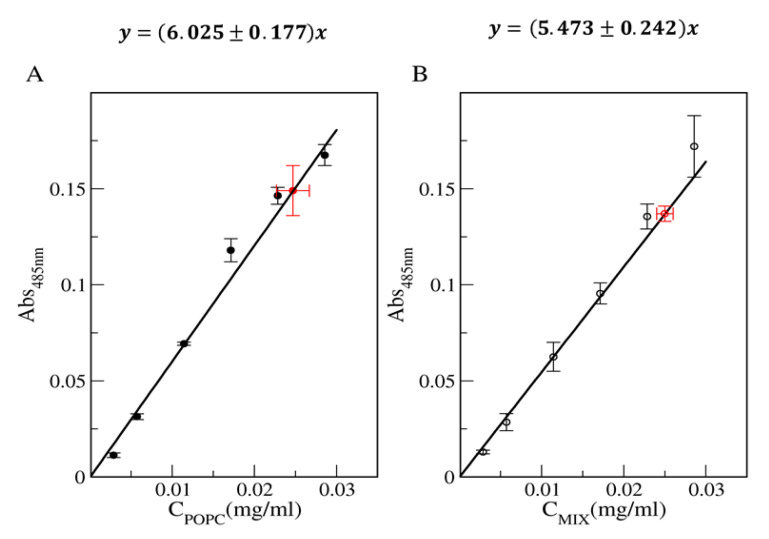
Calibration curves for (**A**) POPC_LIPO and (**B**) MIX_LIPO matrix. The errors’ weighted fitting functions are reported. The red point corresponds to the averaged value ± SE, obtained by a triplicate measurement, respectively for the evaluations of the POPC_LIPO_50 and MIX_LIPO_50 sample concentrations (reported as representative samples), by assuming the calibration curve of the Stewart assay.

**Figure 3 membranes-10-00222-f003:**
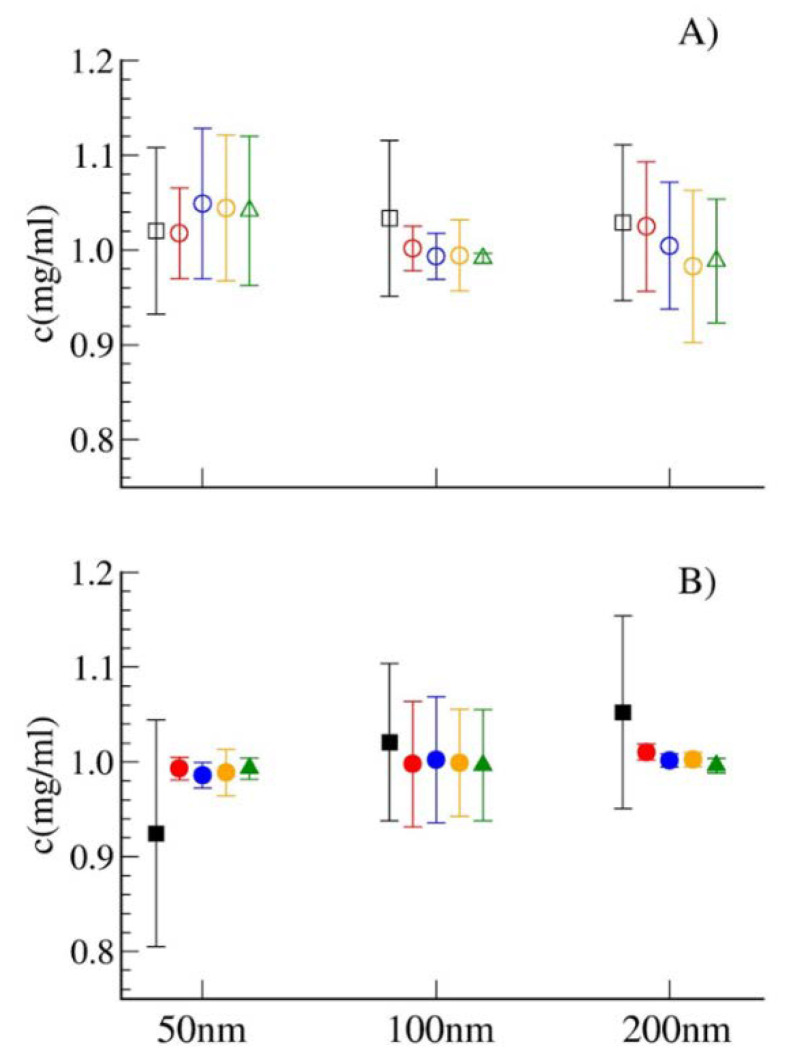
Different-sized vesicles’ weight concentrations. (**A**) POPC_LIPO and (**B**) MIX_LIPO. Stewart Assay (black squares) and the LS-method results, employing both an LS instrument at 60°, 90° and 120° (red, blue and orange circles, respectively) and a Spectrofluorimeter (green triangles).

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
