# Peer review of "Light Scattering as an Easy Tool to Measure Vesicles Weight Concentration"

_membranes, 2020, doi:10.3390/membranes10090222_

Round 1
Reviewer 1 Report
The manuscript entitled “Light Scattering as an easy tool to measure vesicles weight concentration” describes and validates a new method for the quantification of the mass concentration in final liposome dispersions. This is an interesting proposal. Liposomes have emerged as attractive drug carriers as well as membrane models, and in both cases the accurate characterization of vesicle properties seems to be crucial. Aerodynamic diameter and zeta potential are routine controls for liposomes, but mass concentration of final sample is not usually tested. The manuscript is well organized, the aims of the work are clearly described and the experimental procedures adequate. The results are interesting and these are properly analysed and discussed.
According to my opinion, there are some points to be considered for improving the manuscript.
1.- Section 2.2 Preparation of liposomes
Freezing and thawing cycles should be mentioned and described in the section 2.2 Preparation of liposomes. In this section the authors do not mention this step, but the next sentence is read: “To prepare UVs, the obtained MLVs dispersions were extruded through to a mini-Extruder Avestin…”. Then, in the Results and discussion section the following sentence can be read:”In this work, MLVs were prepared by Thin Lay Evaporation (TLE) followed by Freezing and Thawing (FAT) cycles.
2.- The calibration curve used for the Steward Assay were prepared by adding the lipids to the organic solvent, but the liposome samples underwent a two-step procedure. This should be considered and justified. Considering the high error (standard deviation) of results obtained by this method, this issue must be analysed.
3.- The issue 3.1 Theory, in section 3. Results and Discussion might be moved to Section 2 Material and Methods
4.- The issue 2.6. Statistics and errors calculation should be clarified. The uncertainty on the preparation procedure is the same for Steward and LS method, therefore only the uncertainty on the measure should be compared.
5.- Data of aerodynamic diameter and polydispersion index for liposomes extruded through 50, 100 and 200 nm filters should be provided. A table with these results should be added.
Author Response
Please see the attachment
RC

Reviewer 2 Report
In this manuscript, the authors propose the use of light scattering as an effective approach to measure the concentration of liposomal formulations. The authors demonstrate on extruded vesicles of various sizes the ability to use a light scattering instrument with light scattering angles of 60, 90, and 120 to accurately predict the weight concentration of liposomes simply and using conventional bench instruments. While standard assays rely on knowing the composition of lipids present in the formulation of liposomes, this light scattering method has the advantage of being able to accurately measure concentrations from unknown samples. Overall, the work presented herein is of value to those working on the formulation of lipid-based vesicles. Nevertheless, some minor concerns should be considered:
- A disadvantage of some assays (e.g., Bartlett assay) is the requirement of buffers that do not contain inorganic phosphate groups. Does the buffer used for liposome formulation affect the light scattering properties of this LS-based approach?
- Encapsulation of various materials within the liposome core is growing increasingly common as described in Section 1. How does encapsulation of biologics, therapy, or other materials impact the ability to measure concentration following encapsulation? Will this method still be accurate?
- Phosphatidylcholine assays are another common technique used to measure liposome concentration. Perhaps discussion on shortcomings of other assays could be beneficial to the reader.
- The assumption the authors make is that the volume lost during Extrusion I maintains an equal concentration to what is recovered and, therefore, can be subtracted. Could this pose an issue if the lipids lost during extrusion are more concentrated than what is successfully incorporated into liposomes?
Author Response
Please see the attachment
RC
